# Position: Truly Self-Improving Agents Require Intrinsic Metacognitive Learning

Tennison Liu [1]   Mihaela van der Schaar [1]

## Abstract

Self-improving agents aim to continuously acquire new capabilities with minimal supervision. However, current approaches face two key limitations: their self-improvement processes are often rigid, fail to generalize across tasks domains, and struggle to scale with increasing agent capabilities. We argue that effective self-improvement requires intrinsic metacognitive learning, defined as an agent's *intrinsic* ability to actively evaluate, reflect on, and adapt its own learning processes. Drawing inspiration from human metacognition, we introduce a formal framework comprising three components: *metacognitive knowledge* (self-assessment of capabilities, tasks, and learning strategies), *metacognitive planning* (deciding what and how to learn), and *metacognitive evaluation* (reflecting on learning experiences to improve future learning). Analyzing existing self-improving agents, we find they rely predominantly on *extrinsic* metacognitive mechanisms, which are fixed, human-designed loops that limit scalability and adaptability. Examining each component, we contend that many ingredients for intrinsic metacognition are already present. Finally, we explore how to optimally distribute metacognitive responsibilities between humans and agents, and robustly evaluate and improve intrinsic metacognitive learning, key challenges that must be addressed to enable truly sustained, generalized, and aligned self-improvement.

## 1. Introduction

*Metacognition*, colloquially known as "thinking about thinking" or "learning about learning", is a fundamental human ability that enables us to monitor and control our learning processes (Flavell, 1979). At its essence, metacognition

[1]DAMTP, University of Cambridge, Cambridge, UK. Correspondence to: Tennison Liu <tl522@cam.ac.uk>.

*Proceedings of the 42nd International Conference on Machine Learning*, Vancouver, Canada. PMLR 267, 2025. Copyright 2025 by the author(s).

involves applying *metacognitive knowledge*: our understanding of our capabilities, the demands of learning tasks, and the relative merit of different learning strategies. Through this knowledge, we evaluate our learning experiences and refine future plans to optimize learning outcomes.

Metacognition is central to human intelligence and our ability to continuously improve and adapt across diverse environments (Brown, 1987). Consider an athlete learning a new sport: they begin by assessing their current abilities (e.g., strong endurance but limited agility), analyzing the skills required (e.g., ball handling), and identifying transferable training strategies from their previous sport (e.g., mastering fundamentals before advanced techniques). Drawing on this metacognitive knowledge, they develop a structured training plan that progresses from basic to complex drills. As they practice, they monitor performance metrics (e.g., pass success rates) and recognize when to seek coaching as self-practice reaches its limits. Throughout this process, metacognition continuously finetunes learning plans to optimize progress, functioning as an intrinsic process that can operate without external supervision (Cox, 2005).

*Large Language Model* (LLM)-based agents have achieved impressive performance across various domains (Gur et al., 2024; Lu et al., 2024; Wu et al., 2024). These agents are autonomous systems that leverage LLMs as their core decision-making engine, augmented with memory systems and tools for real-world interaction. However, current agents rely on human supervision and environmental feedback that are costly to scale, creating a bottleneck that limits the diversity and depth of their capabilities. This challenge has motivated growing research in *self-improvement* paradigms (Tao et al., 2024), where agents autonomously acquire and learn from experiences with minimal supervision. While self-improvement promises to transcend the limitations of static, data-bound models and mark a shift toward more adaptive learning systems, current approaches face significant constraints. They rely on fixed, human-designed meta-processes that are ill-suited for sustained self-improvement (Zelikman et al., 2022; Song et al., 2024), or constrain learning to narrow domains (Park et al., 2023; Wang et al., 2023a).

**In this position paper, we argue that sustainable, generalized self-improvement requires agents to develop *intrinsic* metacognitive learning abilities.** This intrinsic

capacity allows agents to autonomously and adaptively refine their learning strategies in response to shifting tasks and domains, reducing dependence on human-programmed meta-processes. Drawing from models of human metacognition, we formally introduce a framework for *metacognitive learning* (Section 3), defined as a meta-level process that monitors, evaluates, and regulates a lower-level learning process. Our framework consists of three core components: *(1) metacognitive knowledge*: the ability to assess one's capabilities, understand task demands, and evaluate learning strategies; *(2) metacognitive planning*: the strategic planning of what and how to learn; and *(3) metacognitive evaluation*: ongoing monitoring and reflection on learning progress. Intrinsic metacognitive learning, then, occurs when agents independently assess their learning, update metacognitive knowledge, and adapt learning plans to optimize long-term performance without relying on external mechanisms.

Through this framework, we reinterpret current self-improvement methods as metacognitive processes, where human supervisors assume various metacognitive responsibilities (which we term *extrinsic* metacognition). These responsibilities include determining what to learn (by designing task spaces and acquisition metrics), how to learn (by specifying mechanisms for exploration and experiential learning), and metrics for evaluating self-improvement progress. We identify two scenarios where this fixed, extrinsic design can hamper sustained self-improvement: *domain/-task distribution shift*, where prescribed self-improvement processes may fall short in efficacy, requiring recurring human intervention for continual self-improvement in shifting tasks and domains; *capability-mechanism mismatch*, where fixed metacognitive mechanisms can become increasingly ineffective and misaligned as agent's capabilities evolve.

Through case studies, we explore diverse forms of intrinsic and extrinsic metacognitive learning, observing that self-improvement potential increases when metacognitive functions are more intrinsic yet thoughtfully shared between humans and agents. By analyzing the intrinsic capabilities required for each metacognitive learning component in detail, we show that many essential ingredients are already present in today's LLM agents (Section 5). We conclude by identifying key gaps and open questions for advancing intrinsic metacognitive learning (Section 6). One challenge is developing models of *shared metacognition*, shifting from human-driven extrinsic approaches toward paradigms where metacognitive functions are optimally distributed. Another is evaluating and finetuning intrinsic metacognitive abilities to improve the efficiency and effectiveness of agent self-improvement. Finally, we underscore the need for scalable oversight: as agents autonomously develop capabilities, emergent risks such as unsafe behaviors, misaligned values, and reward hacking increase, demanding oversight mechanisms that evolve alongside agent capabilities.

## 2. Preliminaries

### 2.1. Intelligent Agents

Language model agents (Xi et al., 2023; Yao et al., 2023; Gravitas, 2023; Wang et al., 2024; Hong et al., 2024) are compound systems that harness LLMs (Brown et al., 2020; Ouyang et al., 2022; Chowdhery et al., 2023) as their core computational engine for autonomous reasoning, planning, and action in real-world tasks. While traditional agents relied on handcrafted rules (Wilkins, 2014) or reinforcement learning (Sutton, 1988; Silver et al., 2016)—approaches that struggled to generalize across environments (Lake et al., 2017)—LLM-based agents leverage world knowledge and commonsense understanding acquired through large-scale training to adapt readily to novel tasks. This versatility has enabled applications across diverse domains, from web navigation (Nakano et al., 2021; Gur et al., 2024) and computer interaction (Wu et al., 2024; Xie et al., 2024) to scientific discovery (Lu et al., 2024) and gaming (Wang et al., 2023a).

**Agents.** Following the cognitive architecture proposed in Sumers et al. (2024), an agent's behavior is primarily influenced by its *decision-making policy*, parameterized by the underlying LLM's weights and the agent's source code. The source code contains procedural instructions (commonly known as the system_prompt) that are loaded into working memory at the start of each task. These instructions specify crucial operational aspects, such as using ReAct (Yao et al., 2023) for reasoning and planning, and defining instructions for interfacing with tool APIs (Schick et al., 2023) or memory retrievers (Lewis et al., 2020). The agent's *working memory*, practically realized through the LLM's context, maintains its current working state, effectively a concatenation of all previous states and actions.

**Modules.** Agents can be optionally equipped with two internal modules: a *long-term memory store*, encompassing semantic memory (facts and world knowledge) and episodic memory (the agent's own experiences); and a *tool library*, comprising callable APIs (e.g., a Python interpreter). These modules enhance the agent by providing persistent memory access and enabling real-world interaction through tools. At each decision point, the agent may take a *grounded* action, interacting with the external environment, where the next state is determined by the environment's transition function, or an *internal* action by interacting with its internal modules (e.g., retrieving from memory or calling an API).

### 2.2. Agent Learning Mechanisms

There are two main pathways for learning new behaviors in LLM-based agents: *training-based* methods that directly optimize the weights, and *training-free* techniques that enhance the agent's source code or augment its internal modules. **Training-based.** The LLM's weights encode implicit

procedural knowledge that can be refined through additional training. Prominent approaches include supervised (or imitation) learning from demonstrations (Cobbe et al., 2021; Chung et al., 2024; Zhang et al., 2024b) and reinforcement learning through online exploration (Bai et al., 2024; Havrilla et al., 2024). However, these methods face significant scaling challenges: they are constrained by costly expert annotations or limited to isolated interactive environments with narrow task distributions, typically producing specialist agents with restricted generalization capabilities.

**Training-free.** An alternative approach focuses on optimizing the agent's source code, tools, or memory systems. Prompt optimization techniques (Yang et al., 2024a; Guo et al., 2024) automatically refine procedural prompts to enhance performance. Notable advances by Khattab et al. (2024); Zhuge et al. (2024) conceptualize agent workflows as computational graphs, enabling both optimization of node-level prompts and improvement of overall orchestration (edge connectivity). Agents can also learn from experience by updating their long-term memory: episodic trajectories can be stored in episodic memory (Park et al., 2023; Zhang et al., 2024a) for future retrieval, while new knowledge can be written into semantic memory, including discovered facts (Shinn et al., 2024), scientific insights (Lu et al., 2024), and enhanced tools (e.g., new Python functions) (Wang et al., 2023a; Wu et al., 2024).

**Self-improvement.** The challenges of obtaining human supervision and constructing rich learning environments have motivated research into *self-improving agents* (Tao et al., 2024). The self-improvement process involves meta-level mechanism that autonomously identify learning opportunities and enhance the agent's knowledge and capabilities through experiential learning. This approach substantially reduces the need for manual retraining interventions, while enabling rapid development of sophisticated capabilities and maintaining adaptability to real-world environments. Notable advances include methods that enhance agent reasoning by bootstrapping from self-generated traces (Zelikman et al., 2022; Aksitov et al., 2023) and improve alignment through self-instruction (Bai et al., 2022; Dong et al., 2023). While these approaches eliminate the need for ground-truth labels, they typically require predefined task pools. Recent work has expanded these boundaries by exploring LLMs' capacity to generate their own learning tasks, e.g., for alignment (Wang et al., 2023c) and robotics (Faldor et al., 2024).

## 3. Defining Intrinsic Metacognitive Learning

### 3.1. Formal Framework

We formally frame metacognitive learning as a lens through which to understand self-improvement. Metacognition was first investigated and formalized in developmental psychol-

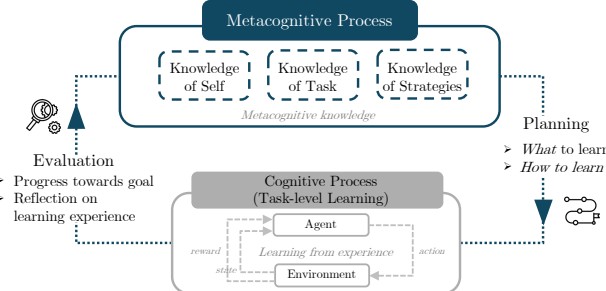

*Figure 1.* **Overview of metacognitive learning.**

ogy, establishing its fundamental role in learning and education (Pintrich, 2002; Zimmerman & Schunk, 2013). Metacognition is a bi-level process in which a *metacognitive* layer monitors, evaluates, and regulates an underlying *cognitive* layer, which could generally entail reasoning, learning, or creative thinking. In our setting of self-improvement, the cognitive-level process is learning itself: engaging with tasks through sensing, acting, and learning from experience (i.e., experiential learning) (Kolb & Kolb, 2009). The metacognitive layer forms a higher-order, closed-loop mechanism that oversees and regulates the learning process to optimize long-term outcomes. More formally, we say:

> **Definition: Metacognitive Learning**
>
> *Metacognitive learning* is a continuous learning process in which a metacognitive system leverages knowledge of learning goals, learning strategies, and agent capabilities (*knowledge*) to plan learning activities for the self-improving agent (*planning*), while continuously evaluating progress and refining future plans (*evaluation*).

We visualize the metacognitive learning process in Figure 1, noting that the definition is neutral to whether the metacognitive process is performed intrinsically by the agent or imposed extrinsically by a human-programmed mechanism.

The process comprises three key components: *metacognitive knowledge*, which includes meta-level understanding of an agent's capabilities (strengths, weaknesses, and existing skills) (Pintrich, 2002), task requirements, and available learning strategies (Serra & Metcalfe, 2009). This knowledge informs the use of *metacognitive skills*: *planning*, which guides resource allocation (what to learn) and learning approaches (how to learn); and *evaluation*, which closes the loop by assessing the effectiveness of the learning plan (Fleur et al., 2021). In Section 5, we examine how these components manifest in current LLM agents and assess the extent to which they demonstrate these capabilities.

### 3.2. Intrinsic *vs.* Extrinsic Metacognitive Learning

At this point, we note that contemporary self-improving agents can be viewed through the lens of metacognitive

learning. However, unlike human metacognition where the process is largely innate, these agents often rely on externally prescribed metacognitive mechanisms designed by human experts to control their learning process. These experts assume metacognitive responsibilities by implementing predefined mechanisms: acquisition metrics for task selection (in curriculum learning (Jiang et al., 2021) and bootstrapped self-training (Singh et al., 2024)), fixed task pools (Li et al., 2024a), static learning strategies (like finetuning (Zelikman et al., 2022) or RAG (Shinn et al., 2024)), and pre-programmed success metrics (Qi et al., 2024). The distinction between *intrinsic* and *extrinsic* metacognitive learning thus reflects the distribution of control of the metacognitive responsibilities: from self-driven adaptation or externally-imposed control of the self-improvement process.[1]

Extrinsic metacognitive learning faces two key challenges: limited adaptability and scalability. First, these mechanisms are typically fixed or externally updated, decoupling them from the agent's learning experiences and evolving capabilities. Second, they do not engage any meta-level reasoning from the learning agent, placing the burden of the metacognitive loop on humans. Consequently, we argue that such processes are insufficient for supporting scalable and generalized self-improvement, especially in two crucial scenarios:

1. **Domain/task distribution shift.** As agents improve, they inevitably encounter new domains and shifting task distributions. Fixed external mechanisms that succeed in one domain (e.g., bootstrapped finetuning for reasoning) may fail in others (e.g., acquiring fine motor skills). For instance, Zelikman et al. (2022); Aksitov et al. (2023); Dong et al. (2023) implement externally designed self-improvement mechanisms that assume particular correctness or task-specific signals. While effective in their intended contexts, these approaches do not generalize easily across domains, creating a bottleneck where further improvement requires recurring human intervention.

2. **Capability-mechanism mismatch.** As agents become more capable, static metacognitive mechanisms can become less effective. These extrinsic systems encode stationary assumptions about the learning process that may not hold as the agent evolves. Recall the athlete learning a new sport: early progress might be driven by simple corrections, but more advanced improvement requires adaptive, nuanced adjustments. As an example, the "generation-verification gap" identified by Song et al. (2024) demonstrated that static self-improvement loops lose efficacy as agents' generative (task-solving) abilities outpace their ability to evaluate their own outputs.

These challenges highlight the limitations of extrinsic metacognitive mechanisms. As an alternative, we advocate for *intrinsic metacognitive learning*, where the agent performs metacognitive functions internally. This activates the agent's meta-level reasoning to adapt its self-improvement processes as learning unfolds. By evolving alongside domain shifts and growing capabilities, such an approach could support more scalable and sustained self-improvement.

### 3.3. Case Studies

To support our position and ground the principles of the metacognitive learning framework, we examine three self-improving agents through detailed case studies. These agents exhibit progressively greater degrees of intrinsic metacognitive learning, illustrating how a shift toward intrinsic processes influences self-improvement dynamics.

**STAR** (Zelikman et al., 2022) showed how reasoning agents can bootstrap their capabilities to tackle increasingly complex tasks. Self-improvement begins with a small set of examples containing both reasoning traces and answers, alongside a larger dataset with answers only. The agent iteratively attempts new tasks and is finetuned on successful traces (those leading to correct answers). In STAR, metacognitive mechanisms are largely extrinsic: the agent lacks active metacognitive knowledge and does not assess its capabilities, understand task characteristics, or consider learning strategies. Task ordering is externally managed by finetuning on problems near the edge of the agent's competence, while learning strategies are guided through rationalization, using answer hints to generate reasoning traces. The agent also does not monitor its learning progress or adapt its self-improvement process. These extrinsic mechanisms are difficult to generalize across task domains, limiting the agent's ability to scale its learning autonomously.

**Voyager** (Wang et al., 2023a) presented an LLM-based embodied agent for lifelong learning in Minecraft. The agent maintains a growing skill library (long-term memory) and continually acquires new capabilities to perform increasingly complex actions. At the cognitive level, it learns new skills represented as executable Python functions. Voyager actively updates its metacognitive knowledge by tracking its capabilities and assessing candidate task characteristics. In planning its learning trajectory, it balances intrinsic goals like exploration and task learnability, and evaluates progress using metrics such as distinct skills acquired and items crafted. This marks a shift toward intrinsic metacognition: both metacognitive knowledge and planning are internalized. Voyager notably outperforms extrinsically guided baselines, including an expert-designed curriculum, underscoring the benefits of agent-directed learning. However, its learning strategy (how it acquires new skills) remains fixed and externally defined. The agent does not reflect on or adapt this

---

[1] We acknowledge a philosophical tension in this definition: if a human programs or instructs an agent to reflect and plan its learning, is the process truly intrinsic? We reconcile this through a *functional autonomy* view, defining the process as intrinsic or extrinsic based on who exercises metacognitive control during learning: the agent or an external mechanism. While all agents are human-bootstrapped, what matters is whether the agent makes the decisions that shape its ongoing learning dynamics.

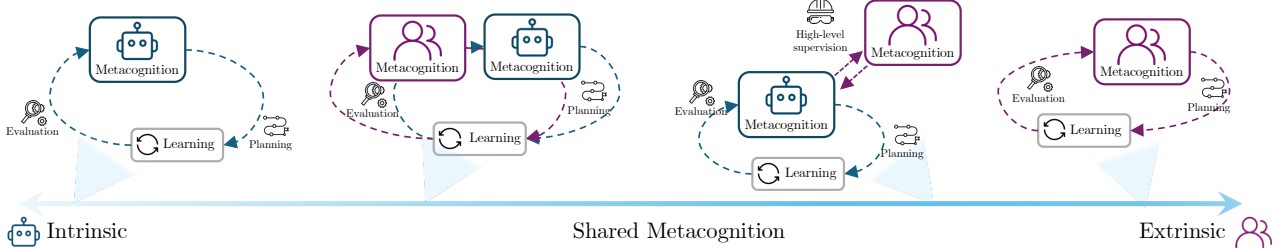

*Figure 2.* **Spectrum of shared metacognition.** From intrinsic, to shared, and externally defined metacognitive processes.

strategy, applying the same exploration-learning loop across tasks. This may limit adaptability as domain complexity or agent capabilities grow (e.g., beyond tool crafting).

While not centered on acquiring problem-solving capabilities, **Generative Agents** (Park et al., 2023) simulate human behavior in a sandbox environment (The Sims), where agents autonomously interact, accumulate experiences, and develop distinct personas. The cognitive-level tasks involve everyday activities such as painting, socializing, or visiting cafés. Agents maintain intrinsic metacognitive knowledge through an evolving understanding of their identities, motivations, and social relationships. They autonomously plan their daily activities, engage in spontaneous interactions, and refine their personalities over time through lived experience. A key metacognitive mechanism in the system is a reflection process. When triggered, agents perform abstract reflection (e.g., recognizing a newfound interest in academic research), updating their semantic memory in ways that shape future planning. This reflection process bears conceptual similarity to self-reflection techniques like Reflexion (Shinn et al., 2024), though with an important distinction. Reflexion operates at the cognitive level, improving performance on specific tasks via feedback on prior attempts. In contrast, Generative Agents reflect at the metacognitive level, reasoning over insights derived from cognitive-level activities to drive long-term behavioral evolution.

These preliminary observations suggest that self-improving agents with intrinsic metacognitive functions achieve more sustained progress, particularly in the diversity and novelty of their evolving capabilities. However, substantiating this hypothesis will require rigorous, systematic studies to more robustly assess the relative strengths and limitations of intrinsic versus extrinsic metacognitive learning.

## 4. Alternative Views

One of the earliest lines of work to explore metacognition in AI systems is metareasoning, which distinguishes between object-level computation and meta-level control, motivated by the need to manage computational costs under resource-bounded rationality (Russell & Wefald, 1991). Metarea-

soning is viewed as a higher-level process that monitors and regulates object-level reasoning or decision-making. A prominent application is in anytime algorithms, where solution quality improves over time and computation can be interrupted at any point (notably, learning can be viewed as an anytime process). For example, Davis (1980) introduced metarules to guide inference in rule-based expert systems, and Genesereth & Smith (1983) used logic programming to reason about which computations to perform. While focused on reasoning, this line of work offers conceptual tools, e.g., resource allocation and decision-making under uncertainty, could prove relevant to learning-centric metacognition.

Recent work has explored metacognition's role in the context of LLM agentic systems, though primarily focusing on its regulation of thinking and problem-solving processes rather than learning and self-improvement processes. Wei et al. (2024) investigated metacognition's potential to enhance AI capabilities across four domains: improving understanding of black-box systems, strengthening reasoning through self-reflection, increasing adaptability via error detection and correction, and refining perception through metacognitive assessment. They proposed neurosymbolic techniques to achieve these capabilities. Similarly, Johnson et al. (2024) examined how metacognition helps agents approach 'intractable' problems: those characterized by ambiguity, uncertainty, novelty, or computational complexity. Their work elucidates the relationship between cognitive-level strategies (e.g., heuristics, analogical reasoning) and metacognitive processes, showing how the latter can regulate the former by seeking additional inputs, selecting appropriate strategies, and dynamically adjusting based on outcomes. While these contributions advance our understanding of metacognition in AI, they complement our position by focusing on task-specific problem-solving rather than the application of metacognition to enhance self-improvement.

Tankelevitch et al. (2024) examined metacognition in human-LLM interactions, focusing on the metacognitive demands placed on human users. Their findings show that effective use of LLMs requires considerable metacognitive effort from users to evaluate and control outputs, motivating the development of support mechanisms such as ex-

plainability and customizability to alleviate this cognitive burden. In a related position, Hughes et al. (2024) argues that *open-endedness*, the capacity to continually generate novel, learnable artifacts, is a core requirement for artificial superintelligence. Viewed through this lens, intrinsic metacognitive learning offers a potential pathway for learning agents to exhibit open-ended evolution.

# 5. Towards Intrinsic Metacognitive Learning

This section analyzes the core components of intrinsic metacognitive learning: metacognitive knowledge, planning, and evaluation. While many examples focus on metacognition in reasoning, the analysis remains relevant, as the only distinction lies in the nature of the cognitive-level task. Whether the agent is reasoning or learning, metacognitive systems operates at a higher level to monitor, regulate, and adapt the underlying process. By assessing the extent to which these intrinsic abilities are present and identifying areas requiring further development, we aim to clarify the path towards realizing intrinsic metacognitive learning.

## 5.1. Metacognitive Knowledge

Intrinsic metacognitive knowledge refers to an agent's ability to assess its own capabilities, understand task demands, and evaluate potential learning strategies. Traditional machine learning agents have not required such abilities, as they operate within fixed frameworks: with predefined datasets (what to learn), algorithms and optimizers (how to learn), and objective functions (learning goals). In contrast, agentic metacognition, where agents actively construct and apply self-knowledge, represents a recent shift toward more adaptive and autonomous learning systems.

The emergence of intrinsic metacognitive knowledge has been enabled by generalist LLM-based agents, which embed world knowledge and exhibit human-*esque*, context-sensitive reasoning and reflection. Recent research has both supported and challenged LLM agents' metacognitive knowledge. Studies have shown that agents can classify mathematical problems based on required solution procedures (Didolkar et al., 2024), demonstrating awareness of their skills and how to apply them. Sachdeva et al. (2024) revealed that LLM agents can identify beneficial training tasks for improving their capabilities, while Wang et al. (2023a); Wu et al. (2024) demonstrated the intrinsic ability to evaluate their current competencies and select learning tasks that would help them acquire new skills.

The most significant challenge to claims of metacognitive knowledge comes from LLM hallucinations, instances where models fail to accurately represent their internal knowledge state (Bender et al., 2021). Other notable limitations include agents' inability to reliably assess their capac-

ity to perform tasks (Kadavath et al., 2022), their confusion about their own capabilities (e.g., access to real-time tools), and their lack of goal awareness (Li et al., 2024b). However, it is worth noting that humans also exhibit systematic errors in self-assessment, such as familiarity bias (Serra & Metcalfe, 2009) and foresight bias (Koriat & Bjork, 2005). In Section 6.2, we discuss some strategies to mitigate hallucinations and enhance metacognitive knowledge.

## 5.2. Metacognitive Skill: Evaluation

Intrinsic metacognitive evaluation encompasses two key aspects: ongoing assessment of learning progress, and reflective analysis of past learning experiences to gauge the effectiveness of strategies and refine future learning approaches.

### 5.2.1. TRACKING PROGRESS

For well-defined capabilities and goals, progress has traditionally been measured through empirical metrics on held-out datasets, using benchmarks for language understanding, mathematical problem-solving, and abstract reasoning (Chollet, 2019; Hendrycks et al., 2021). Self-improvements can then be tracked through improvements in these measurable metrics (Zelikman et al., 2022; Aksitov et al., 2023). However, conventional benchmarks, with their focus on static and narrow task distributions, struggle to evaluate increasingly complex capabilities. While human evaluation offers an alternative (Chiang et al., 2024), it proves similarly time-consuming and difficult to scale. Recent advances have shown that LLM agents can effectively serve as human proxies for intrinsic performance evaluation (Cobbe et al., 2021; Chen et al., 2024a). Notably, LLM-based evaluations excel precisely where quantifiable metrics fall short: assessing difficult-to-measure aspects such as emotional intelligence (Wang et al., 2023b), human preferences (Bai et al., 2022; Zheng et al., 2023), and making fine-grained subjective judgments about creativity (Bradley et al., 2024) and interestingness (Zhang et al., 2024c). This suggests a promising complementarity between external objective metrics and intrinsic evaluation approaches.

### 5.2.2. METACOGNITIVE REFLECTION

This stage of evaluation completes the metacognitive feedback loop by assessing the effectiveness of learning plans and enabling dynamic adjustments to future planning. Unlike task-level reflection, which focuses on improving performance on individual tasks, metacognitive reflection evaluates the agent's overall learning process. It analyzes long-term progress to assess the quality of learning strategies and inform future adaptations. For example, an agent might evaluate how well it allocated effort across tasks with varying characteristics (Wang et al., 2023a; Zhang et al., 2024c), or compare the effectiveness of memory-based learning versus

finetuning. Although cognitive- and metacognitive-level reflection operate at different levels of abstraction, they share underlying mechanisms: assessing performance (e.g., action-level rewards vs. learning progress) and making targeted adjustments (e.g., refining task execution vs. revising learning strategies) (Shinn et al., 2024; Madaan et al., 2024).

Given this, it is worthwhile to ponder the differences between cognitive- and metacognitive-level reflection. A central distinction lies in their *observability*: metacognitive reflection often has access to more complete information about the cognitive process: for example, evaluating a learning plan based on its long-term impact on learning progress. In contrast, cognitive-level reflection typically relies on partial or noisy signals from the environment, such as observations of future states and rewards (Russell & Wefald, 1991). Additionally, because metacognitive reflection operates over representations of the cognitive process, it can, in principle, be *domain-agnostic*. Whether the agent is learning to reason, navigate, or manipulate objects, the structure and function of the metacognitive loop may remain largely consistent, so long as the agent can extract suitable, domain-specific representations of learning progress and outcomes for evaluation.

### 5.3. Metacognitive Skill: Planning

Metacognitive planning is the process by which an agent dynamically refines its learning approaches based on metacognitive evaluations. This involves setting sub-goals, allocating resources, and selecting appropriate learning strategies. At its core, planning revolves around two key considerations: *what* to learn: determining which tasks are most effective for promoting learning (a resource allocation question), and *how* to learn: identifying the most efficient mechanisms for acquiring new capabilities (a strategy question).

#### 5.3.1. WHAT TO LEARN?

The question of what to learn has traditionally been addressed through curriculum learning (Bengio et al., 2009), which assumes a large-scale task pool and seeks to optimize the selection and sequencing of tasks to accelerate knowledge acquisition. In this vein, Dennis et al. (2020); Bauer et al. (2023) introduced expansive, parameterized task distributions where tasks are sequentially sampled based on estimated agent regret, demonstrating sustained skill acquisition. These self-improvement approaches were largely guided by extrinsic planning, with expert-curated task pools and -imposed acquisition methods. More recent advances, however, have shifted control toward the agent itself by leveraging intrinsic drivers such as curiosity and learnability. Approaches like those in Wang et al. (2023a); Zhang et al. (2024c); Lu et al. (2024) use internal notions of exploration, curiosity, and planning to autonomously propose new learning tasks to refine capabilities. Surprisingly, these intrinsic

mechanisms often outperform traditional, human-crafted acquisition formulas, which can mischaracterize learning objectives and introduce unintended biases or inefficiencies.

A major challenge lies in enabling agents to acquire tasks beyond those in pools predefined by human experts. Faldor et al. (2024); Wu et al. (2024) address this by allowing agents to autonomously generate tasks, defining them through Python functions. Another promising direction involves the development of world models that can create arbitrary task environments based on descriptions produced by learning agents (Yang et al., 2024b; Bruce et al., 2024). A separate, yet crucial, challenge lies in obtaining accurate and robust feedback for these generated tasks. Machine learning systems typically rely on well-defined rewards from reliable oracles to guide effective learning, resources that are often unavailable for synthetic tasks. To mitigate this, self-improving agents have demonstrated the ability to learn from partial or synthesized feedback, such as code execution traces (Jiang et al., 2023), or heuristics that identify high-reward actions (e.g., self-consistency (Zelikman et al., 2022)). However, the long-term effectiveness of these approaches remains an open question (Song et al., 2024).

Recent evidence suggests that foundation models can serve as an alternative source of proxy feedback. Techniques such as generating test cases (Chen et al., 2024b), evolving reward functions (Ma et al., 2024), or querying foundation models for estimated rewards (Aksitov et al., 2023; Rocamonde et al., 2024; Chen et al., 2024a) have shown promise. The integration of generalized task generators with comparably general reward models could unlock the full potential of intrinsic metacognition, pushing their self-improvement beyond human-prescribed and finite task sets.

#### 5.3.2. HOW TO LEARN?

At a high level, most learning processes typically consist of two key phases: exploration, where the agent acquires new experiences, and learning, where it extracts knowledge from accumulated experiences. *Exploration.* Various strategies have been developed to enhance exploration. In self-training approaches, a common technique is to modify the agent's policy distribution to promote greater exploration. For instance, temperature scaling is used to encourage sampling of more diverse reasoning paths (Singh et al., 2024; Dong et al., 2023). Other methods introduce uncertainty-guided exploration, such as upper-confidence bound (UCB) techniques for tree search (Hao et al., 2023), iterative exploration driven by self- or LLM-generated feedback (Shinn et al., 2024; Zhao et al., 2024a), and prompt-based instructions designed to encourage agents to seek novel experiences (Lu et al., 2024; Wang et al., 2023a). These exploration strategies aim to increase the likelihood of acquiring experiences that are conducive to acquiring new capabilities, which can

then be reinforced to improve the agent's capabilities.

*Learning.* One form of self-improvement occurs in-weight, where an agent refines its weights through finetuning on self-generated problem-solving traces. This approach has demonstrated significant gains in reasoning performance (Zelikman et al., 2022; Aksitov et al., 2023; Singh et al., 2024), improved alignment with human principles (Wang et al., 2023c; Dong et al., 2023), and facilitated trial-and-error learning in self-generated reinforcement learning (RL) environments (Faldor et al., 2024; Qi et al., 2024). Self-improvement can also take place through long-term memory updates, where agents store and retrieve both positive and negative episodic experiences for future tasks (Zhang et al., 2024a). Additionally, agents can encode abstract decision-making rules or heuristics (Yang et al., 2023; Zhao et al., 2024a) to improve future performance, or accumulate new skills as reusable functions (Wang et al., 2023a; Wu et al., 2024; Zhao et al., 2024b). A critical concern in sustained learning is addressing the risk of *catastrophic forgetting*. As agents continuously acquire new capabilities, they must also preserve core competencies, raising the classic stability-plasticity dilemma: how to balance the integration of new knowledge with the retention of prior expertise (Dohare et al., 2024). This tension is particularly pronounced in long-term, open-ended self-improvement. Existing strategies, such as replay-based methods, which interleave past and present experiences, and divergence-based penalties, which constrain updates to avoid drifting too far from prior policies (Ouyang et al., 2022), offer partial solutions. However, whether such approaches can scale to support sustained self-improvement over time remains an open question.

Despite a growing range of exploration and learning strategies, most self-improving agents still rely on fixed learning mechanisms, applied uniformly across tasks and over time. *Few, if any, systems attempt to adapt or revise how learning occurs in response to changing conditions.* This remains one of the most underdeveloped aspects of intrinsic metacognition. In practice, learning mechanisms are complementary: finetuning refines core reasoning abilities, while memory updates offers a scalable means of accumulating factual knowledge or reusable tools. To succeed in open-ended environments and shifting task distributions, agents must move beyond static learning routines. A truly self-improving agent must not only select effective strategies autonomously, but also continually reflect on and evolve its learning mechanisms to sustain long-term progress.

# 6. Open Questions

## 6.1. Optimal Modes of Shared Metacognition

So far, we have emphasized that some degree of intrinsic metacognitive learning is essential for self-improvement and that intelligent agents already possess many of the necessary capabilities to achieve this. However, an open question remains: *how should metacognitive functions be optimally distributed between agents and humans?* At one extreme, purely intrinsic metacognition grants the agent complete autonomy to define its own improvement goals and regulate its learning progress. At the other extreme, fully extrinsic metacognition relies on continuous human oversight to expand task pools, define acquisition functions, design learning mechanisms, and actively monitor the agent's evolving capabilities. Neither approach is entirely practical. Pure intrinsic metacognition risks the agent becoming trapped in unproductive learning loops, where no meaningful self-improvement occurs. Worse, unchecked self-improvement without external guidance could lead to misalignment with human values, principles, and needs. Conversely, pure extrinsic metacognition is constrained by human limitations, continuous monitoring and control introduce bottlenecks that slow progress and hinder scalability.

This suggests that an effective approach potentially lies in shared metacognition: a balanced framework where both humans and agents share metacognitive responsibilities in evaluation and planning. In Figure 2, we illustrate several possible modes of shared metacognition. *Shared responsibility*: Humans and agents collaboratively oversee learning progress, with metacognitive functions distributed based on their respective strengths. *Hierarchical guidance*: Humans define high-level objectives and constraints, while agents autonomously manage learning and self-improvement within those boundaries. *Gradual handoff*: The system gradually shifts from human-dominated mechanisms to increasing agent autonomy as its intrinsic metacognitive capabilities, trustworthiness, and transparency improve.

## 6.2. Finetuning Intrinsic Metacognition

While much of this discussion has centered on how intrinsic metacognitive capabilities can scale and generalize self-improvement, an equally important question is how these capabilities are themselves developed and finetuned in self-improving agents. For intrinsic metacognitive learning to function effectively, the agent must possess a sufficiently robust metacognitive foundation. As discussed earlier, LLM agents are prone to hallucinations, planning and reflection failures that can undermine core metacognitive functions—such as self-assessment, evaluation, and planning—ultimately impairing their ability to improve learning strategies. Furthermore, as agents evolve, there is no guarantee that their initial metacognitive capacities will remain sufficient or aligned to continuously guide self-improvement.

One approach to this challenge is to mirror the development of metacognitive abilities in humans. Just as students progress from heavily supervised education to increasingly

autonomous graduate research, agents can gradually acquire metacognitive skills. Early development of metacognitive capabilities can rely on human-labeled or curated signals to train or finetune intrinsic metacognition, for example, using reward models to evaluate task selection or learning strategies. This can be complemented by intrinsic metacognitive reflection, where the agent compares learning outcomes across different strategies and adjusts accordingly. These two approaches can be scaffolded: human supervision guides initial metacognitive developments, which the agent then continues to evolve these meta-level skills online during self-improvement. Another promising direction is evolutionary selection (Standish, 2003), where multiple parallel metacognitive learning trajectories are explored in tandem. In this framework, metacognitive capabilities can cross-pollinate and adapt, with selection pressure favoring agents that demonstrate more effective self-improvement.

### 6.3. Evaluating Intrinsic Metacognition

Another important open question pertains to evaluating intrinsic metacognitive capabilities. Reliable evaluation is essential not only for tracking an agent's potential for sustained self-improvement over time, but also as feedback for finetuning metacognitive capabilities. Broadly, we identify three complementary evaluation approaches. The first is an *outcome*-based approach, which assesses the self-improvement achieved by an intrinsic metacognitive learning process: for example, by measuring the rate at which an agent acquires new capabilities or improves on held-out tasks over time. The underlying assumption is that, ceteris paribus, stronger intrinsic metacognition should be observed in more effective self-improvement, though this connection remains indirect. A second, more direct approach is *task*-based: evaluating how efficiently and effectively an agent can learn on a previously unseen probe task. Here, stronger intrinsic metacognitive abilities should enable better self-assessment, planning, and strategy adjustment, leading to faster learning and better overall performance.

A third approach is *component-level* evaluation, which seeks to assess individual metacognitive functions. For example, whether the agent accurately estimates task difficulty (knowledge) or effectively reflects on past learning experiences (evaluation) to identify the most appropriate learning strategy (planning). This often requires *counterfactual* assessment to determine whether better meta-level decisions were available but not chosen, introducing practical challenges. Across all three approaches, a central difficulty is *non-stationarity*: as an agent's capabilities and intrinsic metacognitive functions evolve, the evaluation testbed must adapt accordingly to remain discriminative and relevant. Moreover, effective assessment depends on a high degree of *transparency* or explainability, allowing metacognitive decisions to be interpreted and meaningfully evaluated.

### 6.4. Scalable and Safe Oversight

As metacognitive learning agents grow more autonomous and capable, static or conventional oversight mechanisms will no longer suffice. Left to learn in full autonomy, agents risk developing unsafe behaviors (Zhuang & Hadfield-Menell, 2020), misaligned values (Russell, 2022), or reward hacking (Krakovna et al., 2020). These risks are compounded by a lack of transparency around meta-level decisions that shape the agent's learning trajectory and the capabilities it acquires over time. Addressing these challenges requires oversight that is both scalable and adaptive—evolving in tandem with the agent's growth. A key mechanism is the imposition of safety constraints: hard limits on potentially unsafe behaviors or capabilities that must be continuously evaluated and updated as the agent self-improves. Ensuring value alignment is equally essential, demanding robust monitoring and continued evaluation to confirm that the agent's learning goals remain safe and consistent with human intent. Finally, effective oversight must incorporate robust interpretability tools, enabling humans to trace and understand self-improvement decisions, evaluate emerging capabilities, and intervene when necessary (Nick, 2014).

Another critical challenge arises at the deployment stage. While current agentic systems operate in relatively narrow and controlled environments, general-purpose autonomous agents will likely be deployed in vastly more dynamic, possibly real-world, settings. This shift necessitates large-scale monitoring and rapid intervention mechanisms, not only within controlled sandbox environments but also in physical and digital domains where unintended consequences must be mitigated swiftly. A particularly delicate aspect of real-world deployment is online exploration, where an agent actively evaluates and refines its learning through trial and error. In our view, until robust and continuous monitoring and intervention mechanisms are in place, exploration-based learning should be strictly confined to contained sandboxes.

## 7. Conclusion

Self-improvement is a promising path toward scaling LLM agents to general intelligence. However, current approaches rely on human-designed processes that struggle to adapt to shifting task distributions and increasing agent capabilities. We argue that sustained and generalized self-improvement requires intrinsic metacognition, where agents actively evaluate, plan, and refine their own learning processes. Furthermore, many of the foundational capabilities for metacognitive learning already exist in contemporary agents. Realizing this vision requires advancing shared metacognition, where humans and agents collaboratively manage self-improvement processes, addressing challenges in evaluating and finetuning intrinsic metacognition, and developing scalable oversight mechanisms to ensure safety and alignment.

## Impact Statement

This paper argues that metacognitive learning, the intrinsic ability of self-improving agents to evaluate and regulate their own learning, is essential for scaling intelligence. While this approach holds great promise, it also raises critical concerns about scalable and safe oversight. In Section 6.4, we emphasized these risks and proposed mitigation strategies. However, ensuring alignment, preventing unintended behaviors, and maintaining transparency remain open challenges. Addressing these issues is crucial to realizing the potential of self-improving agents while ensuring their safe and responsible deployment.

## Acknowledgements

We thank the anonymous ICML reviewers, members of the van der Schaar lab, and Andrew Rashbass for many insightful comments and suggestions. TL would like to thank AstraZeneca for their sponsorship and support. This work was supported by Microsoft's Accelerate Foundation Models Academic Research initiative.

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
