# OpenReview forum: "Position: Truly Self-Improving Agents Require Intrinsic Metacognitive Learning"
_ICML.cc/2025/Position_Paper_Track — ICML 2025 Position Paper Track poster_

### Official Review · Reviewer_g9LQ · 2025-03-13

**Significance:** 3
**Argument Clarity:** 3
**Rating:** 4
**Confidence:** 3

**Questions:**

1) What makes a method intrinsic vs extrinsic?
2) If most models, are extrinsic, then can those same models be used for evaluating the metacognitive ability?
3) Does metacognition look different across domains?

**Discussion Potential:**

3

**Paper Summary:**

This paper formalizes a metacognitive framework for viewing self-improving agents. The main position is that it is necessary to have intrinsic metacognitive learning ability, rather than extrinsic metacognitive learning ability, which many works currently have. Intrinsic metacognition does not rely on a human to supervise or describe different metacognitive processes. Common techniques like RAG are considered extrinsic.  Potential limitations of existing work are described as well as examples of existing work displaying aspects of different metacognitive skills in knowledge, evaluation and planning. The paper concludes with open questions in metacongitive learning.

**Position:**

Yes

**Position In Title:**

Yes

**Related Work:**

4

**Strengths And Weaknesses:**

Strengths
- Viewing self-improving agents under a metacognititve framework is interesting and connects well with human intelligence. The sub-types of metacognition are also helpful in terms of understanding strengths and weaknesses of different systems.
- Extensive coverage of different types of agents and connection to different types of metacognition.

Weaknesses:
- It is unclear if the main position is related to intrinsic metacognition or just metacognition? The earlier sections make it seem like it is related to metacognition but the later sections, such as Section 4, discuss many works and it is not clear if those are intrinsic.
- There is little evidence supporting the intrinsic claim. Part of section 3 discusses limitations of existing works but those works were not written for different tasks so it is not necessarily fair to evaluate them in that way.  There is no empirical evidence regarding the distribution shift and capability mechanism mismatch. This is stated on Line 242 which says “based on preliminary observations … not through systematic evaluation”
- It is unclear when a model is considered to have intrinsic metacognitive abilities vs extrinsic metacognitive abilities. Based on the definition on lines 166-167, it seems like almost all methods (as far as I know) have extrinsic metacognitive ability since a human is involved in some way.  More so, some of the works identified in Section 3 as being examples of extrinsic meta-cognition are later referenced in Section 4 as having aspects of intrinsic metacognition. Given the importance of intrinsic vs extrinsic, it should be very clear what is the criteria for the two.

The overall setup, explanation, and description are decent but there is very little evidence to support the claim. The given definition for intrinsic meta-cognition is fairly vague making it hard to evaluate which works have intrinsic vs extrinsic aspects. No metrics are mentioned to measure intrinsic metacognition (not the ability of models to self-evaluate) and there is no evidence showing that some degree of intrinsic metacognition results in better task performance or better generalizability to new tasks.

To improve, it would be better to have a position that can be clearly backed up based on some evidence. A large portion of the paper is devoted to showing that intrinsic metacognition exists. These works provide a perfect opportunity to show that it is better to have intrinsic metacognition vs extrinsic.  It would also be good to include more related work focusing pre-LLM meta-cognitive abilities i.e. meta-learning. Lots of work was done on trying to teach models to “learn to learn.”

**Support:**

3

---

> ### Author Rebuttal · Authors · 2025-03-31
>
> ### [P1] Clarifying position and structure
>
> We appreciate the reviewer’s comment. Our core claim is that achieving sustainable, effective self-improvement will require learning agents to exercise some degree of intrinsic metacognitive learning.
> * In `S3`, we frame metacognition as a lens for understanding self-improvement. We show that many current self-improving systems involve metacognitive processes but with key functions (e.g., task selection or learning mechanisms) imposed by human-designed mechanisms. This motivates our distinction between intrinsic and extrinsic metacognitive learning. We then highlight conditions where primarily extrinsic self-improvement may fall short (further discussed in [P2–3]).
> * `S4` examines whether current LLM agents demonstrate building blocks of intrinsic metacognition. While some examples are drawn from task-solving or reasoning domains, we show they involve components like self-assessment and meta-planning that are transferrable to intrinsic metacognitive learning.
>
> We have revised `S1` and `S3` to clarify and make the position and structure more transparent throughout.
>
> ---
>
> ### [P2] Intrinsic vs extrinsic metacognitive learning
>
> We appreciate the reviewer’s comment and have revised the paper to clarify the definitions and how they apply to the case studies. We now provide the following, more precise definitions:
> * Intrinsic metacognitive learning occurs when the metacognitive processes are controlled, evaluated, and updated by the learning agent itself.
> * Extrinsic metacognitive learning occurs when the agent’s decision space for self-improvement is defined, guided, or constrained by an external agent or mechanism.
> * Shared metacognitive learning occurs when control over metacognitive functions is distributed between the agent and an external system, and may shift over time.
>
> We clarify that among the case studies in `S3.2`, only STAR is purely extrinsic, while Voyager and Generative Agents exhibit components of intrinsic metacognition, such as self-assessment and autonomous planning. We make these distinctions clearer in [this table](https://imgur.com/a/bMxm89B).
>
> ---
>
> ### [P3] Supporting evidence
>
> We appreciate this comment and have further highlighted empirical support for two key claims. First, extrinsic self-improvement processes may fall short under distribution shift or capability-mechanism mismatch and thus require recurring human intervention to remain effective.
> - Distribution shift: Systems like STAR, REST, and RAFT implement extrinsically guided self-improvement, relying on assumptions such as access to correctness signals or task-specific rewards. While effective in their original settings, they are unlikely to generalize to different task domains (e.g., tasks lacking such signals).
> - Capability-mechanism mismatch: Song et al. (2024) showed that self-improvement mechanisms in STAR and REST lose efficacy as the agent's generative capabilities improve, but verification capabilities do not (i.e., reduced efficacy under capability evolution).
>
> Second, we support the claim that a higher degree of intrinsic metacognitive learning leads to more effective self-improvement. We draw on evidence from Voyager and OMNI, two self-improving agents that internally evaluate their learning progress to plan future learning—demonstrating intrinsic metacognitive evaluation and control. Both studies include ablations against extrinsically guided alternatives: OMNI compares to a human-designed learning progress estimator, and Voyager to an expert-defined curriculum. In both cases, agents with intrinsic metacognitive control achieved more efficient and sustained improvement.
>
> ---
>
> ### [P4] Evaluating intrinsic metacognition
>
> Thank you for this comment. In response to your suggestion, we have added a new subsection in `S5` titled “Evaluating Intrinsic Metacognition”. For further detail, please see [P2] of our response to Reviewer `sAN5`.
>
> ---
>
> ### [P5] Relation to Meta-learning
>
> We have added the following text to `S3.3` to connect our position to prior work on meta-learning:
> > A related area that explores “learning to learn” is meta-learning. Meta-learning typically assumes a fixed task distribution and aims to learn an initialization or adaptation mechanism that enables rapid learning on new tasks from that distribution. In contrast, metacognitive learning focuses on self-improvement beyond a fixed distribution—emphasizing intrinsic mechanisms that allow agents to assess their capabilities, reflect on past experiences, and plan future learning. Despite these differences, techniques from meta-learning may still inform the design of intrinsic metacognitive mechanisms—particularly in learning how to select effective learning strategies for new tasks (i.e., metacognitive control) based on meta-learned knowledge from prior learning experiences.
>
> ---
>
> *Thank you again; we hope your concerns have been addressed, and will consider updating their score. We welcome further discussions.*

---

> > ### Comment · Reviewer_g9LQ · 2025-04-07
> >
> > Thank you for the very detailed responses. My initial concerns have all been addressed with the new information. My recommendation for any further revisions is to make as many concrete connections as possible (can be included in the appendix) to the different definitions to better help the reader. The new comments are perfect examples of this.
> >
> > I have updated the score to 4.

---

> > > ### Author Response · Authors · 2025-04-08
> > >
> > > Thank you, we are glad we could address your concerns and appreciate your constructive feedback! We will incorporate the recommended revisions to further improve our work.
> > >
> > > The Authors of #354

---

### Official Review · Reviewer_sAN5 · 2025-03-13

**Significance:** 4
**Argument Clarity:** 3
**Rating:** 4
**Confidence:** 4

**Questions:**

- I don't quite get the point of *shared* metacognition where it seems not central to the Position. Maybe explain in more details how it is important and how it differs from just fine-tuning metacognition.

- The Position has been light on how to measure the metacognition abilities which is important but very hard to achieve, I would like to see some insights on how might be solutions on the measures/metrics.

**Discussion Potential:**

3

**Paper Summary:**

The paper identifies the importance of metacognitive learning, in which the agent can intrinsically evaluate, reflect, and adapt their learning. It divides metacognitive abilities into three components: metacognitive knowledge, metacognitive planning, and metacognitive evaluation. It discusses the option questions on how to train and fine-tune metacognition, how to achieve robust improvement, etc. It points out that these abilities exist in the current agents, but the shared metacognition requires investigation.

**Position:**

Yes

**Position In Title:**

Yes

**Related Work:**

3

**Strengths And Weaknesses:**

Strengths
+ They identify the importance of intrinsic metacognitive learning abilities for self-improving LLMs where the learning process has to change dynamically; and they give a clear definition of metacognitive learning;
+ They ground their reinterpretation to the current agents as case studies like STAR and Voyager;
+ They discuss the three components Metacognitive knowledge,  Metacognitive Skill: Evaluation, Metacognitive Skill: Planning in detail on what they are, how they help agents self-improve and how they can be learned.

Weaknesses
- I don't quite get the point of *shared* metacognition where it seems not central to the Position. Maybe explain in more details how it is important and how it differs from just fine-tuning metacognition.

**Support:**

4

---

> ### Author Rebuttal · Authors · 2025-03-31
>
> *We appreciate the reviewer’s thoughtful evaluation and positive feedback.*
>
> ---
>
> ### [P1] Shared metacognition vs finetuning metacognition
>
> Thank you for this comment. If we understand correctly, your question concerns the distinction between shared metacognition (`S5.1`) and finetuning intrinsic metacognitive abilities (`S5.2`), and how both relate to the central position.
>
> **Central position.** Our core claim is that sustainable and generalizable self-improvement requires learning agents to exercise some degree of **intrinsic metacognitive learning**—the ability to plan, monitor, and evaluate their learning processes. Shared metacognition addresses *how* these metacognitive functions are distributed between the agent and external systems or humans, forming a spectrum between purely intrinsic and purely extrinsic processes (as illustrated in `Fig 2`).
>
> **S5.1 vs S5.2.** In `S5.1`, we argue that relying solely on either extreme is likely insufficient: fully intrinsic agents risk drifting into ineffective or misaligned learning behaviors, while fully extrinsic processes create bottlenecks through constant human supervision. Shared metacognition offers a scalable middle ground, where metacognitive responsibilities are shared and may shift dynamically over time. In contrast, `S5.2` focuses on how intrinsic metacognitive abilities can be improved over time—for example, learning better ways to plan or evaluate their learning trajectories. This is crucial for adapting to growing capabilities and mitigating capability-mechanism mismatch.
>
> We hope this clarifies the relevance of shared metacognition and how it differs from simply fine-tuning metacognitive functions. Please let us know if you’d like to see any of this elaborated further.
>
>
> ---
>
> ### [P2] Evaluating intrinsic metacognitive functions
>
> We appreciate this suggestion. We agree that evaluating intrinsic metacognitive abilities is both essential and challenging. In response, we have added a new subsection titled **“Evaluating Intrinsic Metacognition”** (new `S5.3`) to the revised manuscript, where we outline the importance of intrinsic metacognition assessment, several possible evaluation strategies, and discuss associated challenges. Please see the added subsection below.
>
>
> > Another important open question pertains to evaluating intrinsic metacognitive abilities. Robust and accurate evaluation is essential not only for tracking an agent's self-improvement over time, but also for comparing and finetuning intrinsic metacognitive functions to ensure efficient self-improvement. Broadly, we identify three complementary evaluation approaches. The first is an *outcome*-based approach, which assesses the outcomes produced by intrinsic metacognitive learning—for example, by measuring the rate at which an agent acquires new capabilities or improves on held-out tasks over time. The underlying assumption is that, ceteris paribus, stronger intrinsic metacognition should be observed in more effective self-improvement, though this connection remains indirect. A second, more direct approach is *task*-based: evaluating how quickly and effectively an agent can learn a previously unseen task. Here, stronger intrinsic metacognitive abilities should enable better self-assessment, planning, and strategy adjustment, leading to faster and more effective task learning. A third approach is *component-level* evaluation, which aims to assess individual metacognitive functions—for instance, whether the agent accurately estimates task difficulty (knowledge) or selects the most suitable learning strategy (planning).  This often requires **counterfactual** reasoning to determine whether better options were available (but not chosen), posing practical challenges. Across all three approaches, a key difficulty is **non-stationarity**: as an agent’s capabilities and metacognitive functions evolve, evaluation frameworks must adapt accordingly. Additionally, effective assessment depends on a high degree of **transparency** or explainability, so that metacognitive decisions can be interpreted and meaningfully assessed.
>
> ---
>
> *Thank you again; we hope your concerns have been addressed, and you will consider updating their score. We welcome further discussions.*

---

### Official Review · Reviewer_Dsek · 2025-03-13

**Significance:** 4
**Argument Clarity:** 4
**Rating:** 5
**Confidence:** 4

**Questions:**

Please see main review

**Discussion Potential:**

4

**Paper Summary:**

Modern LLM-based agents have the ability to improve their own behavior through targeted learning.  But what should they learn and how should they learn it?  Currently humans provide a kind of external "metacognitive mechanism" that makes these choices.  But as systems are deployed in more complex and changing environments, it will be unreasonably to imagine humans "following them around" and telling them what to learn.  Instead, human engineers should focus on developing and improving *intrinsic* metacognitive methods that will enable agents to adapt the ways in which they learn about the world over time.

## update after rebuttal
I remain very positive about this paper

**Position:**

Yes

**Position In Title:**

Yes

**Related Work:**

4

**Strengths And Weaknesses:**

This paper is beautifully clear and well written.  The position is stated clearly and well argued.  The authors are not promoting their own work and do a very nice job of framing related work and educating the reader about the area.

I do think the setting could be made a bit clearer, still, though.  I think it's useful to be very clear about what aspect of metacognition we are talking about, when.  Here is one way to think about it:
A. Reasoning about how to control one's reasoning
B. Reasoning about how to control one's learning
C. Learning about how to control one's reasoning
D. Learning about how to control one's learning
As far as I can tell, this paper focuses only on B and D, with (I think) an emphasis on D.  But, really, all of these settings may have some role to play in a very general-purpose intelligent agent.  Section 3.3 goes into this distinction a bit, but it felt like a bit too little, a bit too late for me.

Older literature in metacognition (btw, the term may be due to Flavell, 76 but I didn't dig back into the literature to verify that), such as work by Russell and Wefald and lots of later work on anytime algorithms, or deciding when to quit trying to solve a particular planning problem (because it's only semi-decidable) has focused on cases A and C, but there may be things to learn from it.

Another way to be clearer about your enterprise would be to remember that there are lots of kinds of "agents" in AI, from McCarthy's thermostat, to self-improving robots, to the LLM-based agents you focus on.  I think it's important to either:
- initially, make clear that this paper is about LLM-based agents (and, ideally, indicate that in the position and titile)
- broaden your discussion and talk about other types of agents and what role metacognition has for them
One could argue that learning the value-function for chess playing is a kind of type C meta-cognition.  Or look at some relevant literature situated in robotics (e.g. https://arxiv.org/abs/2402.15025 or https://arxiv.org/abs/2101.03834).

One more thing to address is time-pressure.  At least in cases A and C, where the goal of metacognition is probably to improve the quality of reasoning results as a function of time, it's critical to observe that meta-reasoning happens *in time* and therefore has to be worth the time it spends.  T

Here are some other thoughts:
- If "intrinsic" is important, put it in the title!
- A crazy connection to ancient literature is Brian Cantwell Smith's PhD thesis on "reflection" in computer systems (e.g. "Reflection and Semantics in LISP").  The idea of enabling LLMs to reflect on their own internal processes is probably crucial to this enterprise.  CoT is sort of this, but also sort of not, I think.
- A writing mistake?  "Until recently, the possibility for AI systems to possess such knowledge was intractable."
- "benchmarks...struggle to evaluate"  I personally don't think benchmarks can struggle!  Or, really, evaluate

**Support:**

4

---

> ### Author Rebuttal · Authors · 2025-03-31
>
> *We appreciate the reviewer’s thoughtful evaluation and positive feedback.*
>
> ---
>
> ### [P1] Clarifying scope
>
> Thank you for this insightful comment. We agree that clarifying the scope of metacognition under discussion is important, and we found your A–D typology helpful for framing this distinction. Building on this lens, we view metacognition as a bi-level process in which a metacognitive system monitors, evaluates, and regulates an underlying cognitive-level process—which could be reasoning, creative thinking, or learning. Types A and C correspond to metacognition applied to *reasoning*, while Types B and D (our focus in this paper) correspond to metacognition applied to *learning*.
>
> Our emphasis is primarily on Types B/D, which is an agent's capacity to monitor and improve its own learning processes over time to support self-improvement. That said, prior work on Types A and C—such as Reflexion, RAP, and classical meta-reasoning frameworks (e.g., Russell & Wefald)—offer valuable foundations. For example, Reflexion and RAP use metalevel processes to guide reasoning via intrinsic reflective mechanisms or extrinsic search-based planning. More recent systems like DeepSeek R1 and OpenAI’s o1 aim to learn how to reason more effectively. While these types differ in the nature of the cognitive process (reasoning vs. learning), they are structurally aligned. In `S4`, we analyzed systems developed for Type A/C settings and argued that the intrinsic abilities they demonstrate (e.g., self-evaluation, planning, and adaptation) are potentially transferable and could support metacognitive learning of Type B/D.
>
> In response to your comment, we have revised `S1` and `S3` to make this framing and the distinction between different cognitive-level processes more explicit earlier in the paper.
>
> ---
>
>
> ### [P2] Connection to Metareasoning and (time/resource)-bounded optimality
>
> Thank you for highlighting the connection to earlier work on metareasoning and time-sensitive control of computation. We agree that there is a strong conceptual parallel between this literature and our framing of metacognitive learning. Classic metareasoning frameworks—such as those by Russell and Wefald—distinguish between object-level computation and meta-level control, motivated by the need to manage computational costs under resource-bounded rationality. In these settings, decision-theoretic metareasoning approaches are used to determine whether continued computation is worth the time or resource cost.
>
> We see similar challenges arising in metacognitive learning. Learning itself can be viewed as an *anytime* process: more learning typically improves performance, but the returns are unpredictable and rarely provably complete. Like reasoning, learning is subject to time and resource constraints, and metacognitive control must manage this uncertainty. In this context, metacognitive learning addresses questions such as *Given limited resources, which learning tasks should be prioritized, and how should they be approached?* This includes allocating effort across tasks and learning strategies and monitoring outcomes to guide future learning.
>
> To better reflect this connection, we have added the following related position in `S3.3` of the revised manuscript:
>
> > One of the earliest lines of work to explore metacognition in AI systems is metareasoning, which distinguishes between object-level computation and meta-level control, motivated by the need to manage computational costs under resource-bounded rationality \citep{russell1991principles}. Metareasoning is viewed as a higher-level process that monitors and regulates object-level reasoning or decision-making. A prominent application is in anytime algorithms, where solution quality improves over time and computation can be interrupted at any point (notably, learning can be viewed as an anytime process). For example, \citet{davis1980meta} introduced metarules to guide inference in rule-based expert systems, and \citet{genesereth1983overview} used logic programming to reason about which computations to perform. While focused on reasoning, this line of work offers conceptual tools—such as resource allocation and decision-making under uncertainty—that could prove relevant to learning-centric metacognition.
>
> ---
>
> ### [P3] Other comments
>
> * **Clearer positioning.** Following your suggestion, we have adjusted the title to 'Self-Improving **LLM Agents** Require **Intrinsic** Metacognitive Learning', and made clear we are referencing LLM-based agents in `S1`.
> * **Reflection.** We have referenced Cantwell Smith's work on Reflection in `S4.2`, which offers useful conceptual grounding for self-reflective capabilities in LLM agents.
> * **Typos.** The phrasing and grammatical issues have been rectified. Thanks!
>
> ---
> *Thank you again for your thoughtful review; we hope your comments have been addressed. We welcome further discussions.*

---

### Decision · Program_Chairs · 2025-04-30

**Decision:**

Accept (poster)

**Comment:**

Reviewers loved the topic and felt the paper was well written with clear introduction.
Good section on open Questions discusses issues and nuances.
Good section on case studies.
One reviewer pointed out the area of "learning to learn" is relevant.
Reviewers suggest "intrinsic" should be in the title.

The alternative views should be identified explicitly in an "Alternative Views" section, as instructed in the CFP.  Some alternative views are discussed under Related Work, and varieties and degrees of meta-cognition under Open Questions.  Open Questions should be rephrased as Alternative Views.